# ESPVR: Entity Spans Position Visual Regions for Multimodal Named Entity Recognition

**Xiujiao Li   Guanglu Sun*   Xinyu Liu**
Harbin University of Science and Technology, Harbin, China
2120410084@stu.hrbust.edu.cn   *corresponding author: sunguanglu@hrbust.edu.cn
2010400001@stu.hrbust.edu.cn

## Abstract

Multimodal Named Entity Recognition (MNER) uses visual information to improve the performance of text-only Named Entity Recognition (NER). However, existing methods for acquiring local visual information suffer from certain limitations: (1) using an attention-based method to extract visual regions related to the text from visual regions obtained through convolutional architectures (e.g., ResNet), attention is distracted by the entire image, rather than being fully focused on the visual regions most relevant to the text; (2) using an object detection-based (e.g., Mask R-CNN) method to detect visual object regions related to the text, object detection has a limited range of recognition categories. Moreover, the visual regions obtained by object detection may not correspond to the entities in the text. In summary, the goal of these methods is not to extract the most relevant visual regions for the entities in the text. The visual regions obtained by these methods may be redundant or insufficient for the entities in the text. In this paper, we propose an Entity Spans Position Visual Regions (ESPVR) module to obtain the most relevant visual regions corresponding to the entities in the text. Experiments show that our proposed approach can achieve the SOTA on Twitter-2017 and competitive results on Twitter-2015.

## 1   Introduction

Named entity recognition (NER) is a fundamental task in the field of information extraction, which can automatically recognize named entities in text and classify them into predefined categories. NER has been widely used for many downstream tasks, such as entity linking and relationship extraction. With the rapid development of social media, multimodal deep learning is widely used to perform structured extraction from massive multimedia news and web product information. Among them, Multimodal Named Entity Recognition (MNER)

aims to identify and classify named entities from text using images as auxiliary information. MNER can disambiguate multi-sense words by augmenting linguistic representations with visual information, resulting in superior performance compared to traditional Named Entity Recognition (NER).

While previous efforts have yielded promising results, they still fall short in effectively selecting visual information. For existing methods of utilizing visual information, we classify it into two types: global visual information and local visual information.

Some previous works (Lu et al., 2018; Zhang et al., 2018; Yu et al., 2020; Chen et al., 2021; Sun et al., 2021, 2020; Liu et al., 2022a,b; Wang et al., 2022) consider that if the whole image information is input to the multimodal interaction module, then such image information is global visual information. However, the multimodal interaction module relies on attention to select the visual regions associated with the text for interaction. Therefore, the visual information that eventually interacts with the text is mainly the local visual information related to the text. In other words, even if the input to the multimodal interaction module is the whole image information, finally the local visual information is selected using an attention-based method and then interacts with the text, so we consider it as a process of using attention to select the local visual information. When using attention to extract visual regions, attention is distracted by the entire image, rather than being fully focused on the visual regions most relevant to the text. Therefore, using an attention-based method to select the local visual information not only obtains valuable visual information but also introduces irrelevant visual information.

Besides, most of the previous approaches (Wu et al., 2020; Wang et al., 2020; Zheng et al., 2020; Zhang et al., 2021; Wang et al., 2021) use object detection (e.g., Mask R-CNN) to detect visual ob-

ject regions, and treat visual objects as local visual information to interact with the text. However, object detection has a limited range of recognition categories, so it may not detect all objects within the categories defined by the dataset. Moreover, the visual regions obtained by object detection may not correspond to the entities in the text.

In summary, the goal of these methods is not to extract the most relevant visual regions for the entities in the text. The visual regions obtained by the above two methods may be redundant or insufficient for entities contained in the text, leading to identifying a non-entity as an entity or incorrectly predicting an entity category. Therefore, to obtain the most relevant visual regions for the entities in the text, we propose an Entity Spans Position Visual Regions (ESPVR) module. Specifically, the ESPVR module consists of two modules: Entity Spans Identifying (ESI) module, Visual Regions Positioning (VRP) module. First, the ESI module identifies all entity spans in the text. Then the VRP module uses these entity spans to extract entity features and uses the entity features to locate the visual regions that are most relevant to the entities in the text.

To summarize, the major contributions of our paper are as follows:

• We propose a novel ESPVR module for MNER, which can select the most relevant visual regions for the entities in the text. The ESPVR module consists of two modules: Entity Spans Identifying (ESI) module, and Visual Regions Positioning (VRP) module.

• We conduct extensive experiments on two benchmark datasets, Twitter-2015 and Twitter-2017, to evaluate the performance of our ESPVR module. Experimental results show that the ESPVR module outperforms the current state-of-the-art models on Twitter-2017 and yields competitive results on Twitter-2015.

## 2 Related work

In general, studies about MNER are similar in terms of text feature extraction. However, there are differences in research methods when using image information and fusing modal information. The existing work can be classified into the following two categories based on the use of visual information:

(1) The entire image is equally segmented into multiple visual regions by the convolutional architecture (e.g., ResNet), and then using the multimodal interaction module with attention to select the visual regions associated with the text.

In fact, not all visual regions within an image are beneficial to improve the accuracy of the model prediction. To address this problem, some researchers proposed a method of first dividing the whole image into multiple visual regions equally, and then extracting the most relevant visual regions to the text in order to filter out irrelevant visual information. The visual information that ends up interacting with the text is actually the local visual information after filtering, even if the input is global visual information. Lu et al. (2018) used the pre-training ResNet model to extract visual regions and then added them to the text embedding by a visual attention model. To make full use of text and visual information, Zhang et al. (2018) used adaptive co-attentive networks to fuse text embedding and visual regions representation. Yu et al. (2020) proposed a multimodal interaction module to obtain both image-aware word representations and word-aware visual representations, and used text-only entity spans detection as an auxiliary module to mitigate visual bias. Chen et al. (2021) used an external knowledge database to obtain the final multimodal representation by attention-guided visual layer.

Irrelevant text-image pairs account for a large proportion of the dataset. Therefore, Sun et al. (2021) and Sun et al. (2020) used a modified BERT encoder to obtain information for inter-modal fusion and then introduced text-image relationship classification as a subtask to determine whether image features were useful. In addition, Liu et al. (2022a) proposed a novel uncertainty-aware framework for social media MNER.

In addition, for the problem of fine-grained semantic correspondence between objects in images and words in the text. Liu et al. (2022b) performed an enhanced representation of each word in the text by semantic enhancement and performed cross-modal semantic interaction between text and vision at different visual granularities. Wang et al. (2022) proposed a Scene graph driven Multimodal Multi-granularity Multitask learning framework.

(2) Obtaining visual objects from the whole image by object detection (e.g., Mask R-CNN), and treating visual objects as local visual information to interact with the text.

Visual objects are considered fine-grained image

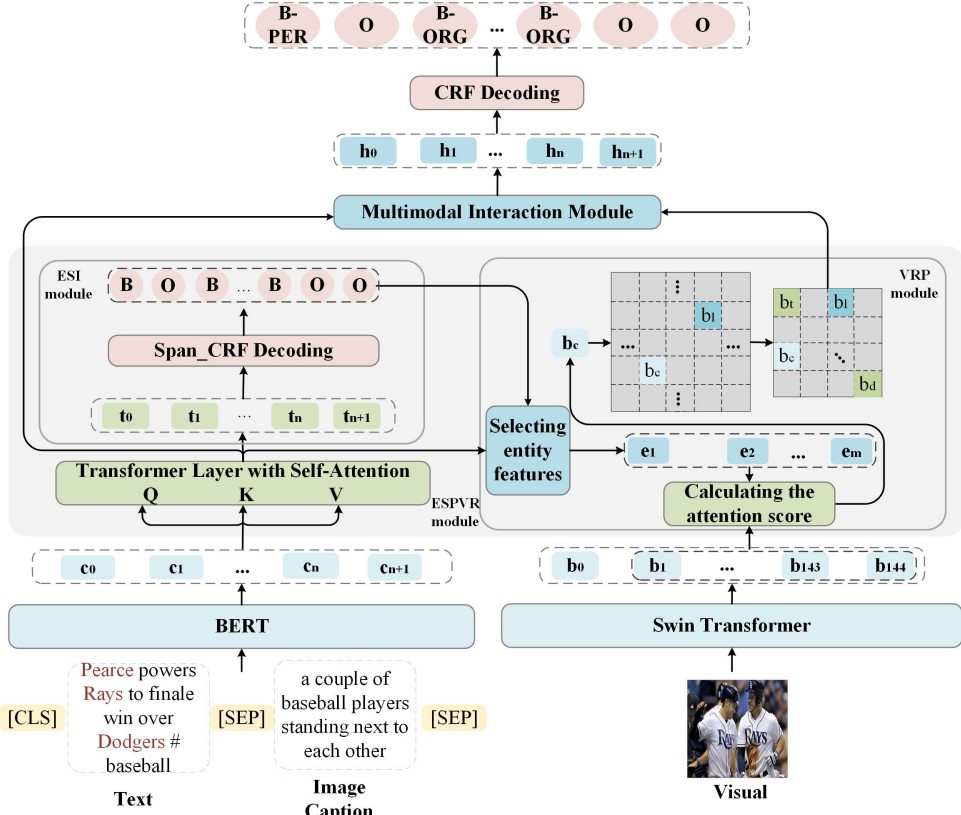

Figure 1: Overall Architecture of Our ESPVR.

representations, and for the text with multiple entity types, the visual object regions of the related images can be used to capture different entity information. Wu et al. (2020) used Mask RCNN for object detection, they embedded top-k objects into vectors to interact with text features through the dense co-attentive module. In addition to this, Wang et al. (2020) proposed a multimodal alignment framework that used a contrastive objective, to guide alignment between visual and textual representations. Zheng et al. (2020) used an adversarial learning technique that aimed to fuse textual and visual features into a common feature space. For the same problem, Zhang et al. (2021) first extracted noun phrases in the text and visual object regions in images and then used GNN to model the relations between them.

Although many multimodal neural techniques have been proposed to incorporate images into the MNER task, the ability of models to exploit multimodal interactions remains unclear. Therefore, Chen et al. (2020) provided an in-depth analysis of existing multimodal fusion techniques from different perspectives and described a situation where the use of image information did not always im-

prove performance. Based on this work, Wang et al. (2021) aligned image features into the text space by using image-text alignment to better utilize the attention mechanism in transformer-based pre-trained textual embeddings.

## 3  Method

In this section, we first introduce the task definition, and then describe the proposed model in detail.

**Task Formulation**: Given a text and image pair $(X; V)$ as input, where $X = \{x_1, ..., x_n\}$ and $n$ denotes the length of the text, the goal of MNER is to extract a set of entities from $X$ and classify each entity into one of the pre-defined categories with the assistance of image information, e.g., Person (PER), Location (LOC), Organization (ORG), and Miscellaneous (MISC). As with most existing work in MNER, we regard the task as a sequence labeling problem. Specifically, let $Y = \{y_1, y_2, ..., y_n\}$ represent a label sequence corresponding to $X$, where $y_i \in \zeta$ and $\zeta$ is the pre-defined label set with standard BIO2 tagging schema.

## 3.1 Overall Architecture

Fig. 1 illustrates the overall architecture of the ES-PVR, which consists of four major modules: (1) Feature Extraction module; (2) Entity Spans Position Visual Regions (ESPVR) module; (3) Multimodal Interaction module; (4) CRF Decoding module.

We first obtain word representations and visual representations, respectively. Then, to obtain local visual information, we deploy Entity Spans Position Visual Regions (ESPVR) module to position the visual regions that are most relevant to the entities in the text. Next, a Multimodal Interaction module is devised to fully capture cross-modality semantic interaction between textual hidden representation and visual regions hidden representation. Finally, the CRF Decoding module assigns an entity label to each word in the input sequence, leveraging the hidden representations obtained from the Multimodal Interaction module.

## 3.2 Feature Extraction Module

**Word Representations**: To better model the semantic information of text X and get different representations for the same word in different contexts, we leverage pre-trained language model BERT (Devlin et al., 2018) as our text encoder. Moreover, image captions from an image captioning model can fully describe the whole image and provide more semantic information. Therefore, to let the text learn contextual information from an image caption, we first use $[SEP]$ to concatenate the text and image caption as a cross-modal input view rather than the text-only input view. Then, we denote the input as $X\prime = \{[CLS], x_1, x_2, ..., x_n, [SEP], x\prime_1, x\prime_2, ..., x\prime_{n'}, [SEP]\}$, where $x_i$ is the word of the text, $x_i\prime$ is the word of the image caption, $[CLS]$ and $[SEP]$ are special tokens of BERT. Lastly, the text $X\prime$ is fed into the BERT to get the word contextualized representations $C = \{c_0, c_1, c_2, ..., c_{n+1}\}$ is the generated contextualized representation for $x_i$, $c_i \in \mathbb{R}^d$, $d$ stands for the dimension of the word embedding.

**Visual Representations**: To obtain better visual representations, we use the pre-trained model Swin Transformer (Liu et al., 2021). Specifically, given an image $V$, the visual representations $B = \{b_0, b_1, ..., b_{144}\}$ are obtained by extracting the output of the last layer of Swin Transformer, where $b_0$ represents the feature of the whole image, usually used for image classification, $\{b_1, ..., b_{144}\}$ are the $12 \times 12 = 144$ visual region features divided by Swin Transformer, and each region is represented by a 1536-dimensional vector.

## 3.3 Entity Spans Position Visual Regions (ESPVR) Module

As shown on the left side of the ESPVR module in Fig. 1, we first use a transformer layer with self-attention (Vaswani et al., 2017) to capture the intra-modality relation for the text modality and obtain each word's textual hidden representation $T = \{t_0, t_1, ..., t_{n+1}\}$, where $t_i \in \mathbb{R}^d$ denotes the generated hidden representation for $x_i$.

**Entity Spans Identifying (ESI) module**: The purpose of the EBI is to identify the position of the head and tail of the entities in the text, which can be used for positioning the visual regions that are most relevant to the entities in the text.

We remove the type information and define the set of span labels $Z' = \{B, I, O\}$, and use $Z = \{z_1, ...z_n\}$ to denote the sequence of labels, where $z_i \in Z'$. Subsequently, $T$ is fed into the Span_CRF decoding layer to predict a sequence $Z$ of labels of $X$.

**Visual Regions Positioning (VRP) Module**: After obtaining the sequence $Z$ of labels of $X$, we first need to select entity features $E = \{e_1, e_2, ..., e_m\}$ corresponding to entities from $T$ based on the labels in $Z$, where $e_i$ stands for the feature of the $i-th$ entity, $m$ stands for the number of entities included in the text. Then, to maintain the same scale as most of the original images, we extract a visual region feature $v$ of size $\alpha \times \alpha$ that is most relevant to the entities in the text. The specific process is as follows:

First, we select a visual region feature $b_c$ from $\{b_1, ..., b_{144}\}$ that is most relevant to the entities in the text. Specifically, we take each entity feature in $E = \{e_1, e_2, ..., e_m\}$ as $Q$, each visual region feature in $\{b_1, ..., b_{144}\}$ as $K$, and calculate the correlation score $F_{(e_i, b_j)}$ between each visual region feature and each entity feature, and sum up the correlation scores of each visual region feature and all entity features to obtain 144 correlation scores. We select a visual region feature $b_c$ corresponding to the maximum value from the sum of 144 correlation scores:

$$F_{(e_i, b_j)} = softmax(\frac{QK^T}{\sqrt{d_K}}) \qquad (1)$$

$$b_c = \underset{1 \leq j \leq 144}{\arg \max}(\textstyle\sum_{i=1}^{m} F_{(e_i, b_j)}) \qquad (2)$$

where $d_k$ is the dimension of key vector $K$.

Then, we select neighboring visual region features based on the index $c$ of $b_c$. Specifically, the positions of all visual region features are expressed in row and column coordinates, and visual region features with row and column distances less than $\alpha$ are identified as neighboring visual region features of visual region feature $b_c$, where the range of $\alpha$ is $1 \leq \alpha \leq 11$. Because the number of all visual regions is $12 \times 12$, the upper limit of $\alpha$ is 11.

Next, we select a neighboring visual region feature $b_l$ that is most relevant to $b_c$ from all neighboring visual region features. Specifically, we take the visual region feature $b_c$ as $Q$ and each neighboring visual region feature as $K$, use attention to calculate the correlation score between $b_c$ and each neighboring visual region feature, and select a neighboring visual region feature $b_l$ with the largest correlation score.

Finally, to maintain the same scale as most of the original images, we extract a visual region feature $v$ of size $\alpha \times \alpha$. Specifically, we compare the coordinate of the visual region feature $b_c$ and the coordinate of the neighboring visual region feature $b_l$, and select a minimum number of rows and columns as the coordinates of the top-left visual region feature $b_t$ for the visual region feature $v$. And using $\alpha$ as the edge length of $v$, we add $\alpha$ to the rows and columns of coordinate $t$ to get the coordinate $d$ of the down-right visual region feature $b_d$ for visual region feature $v$. We use all the visual region features that lie within the range of $t$ and $d$ to form an overall visual region feature $v$.

### 3.4 Multimodal Interaction Module

Following (Yu et al., 2020), we stack the cross-modality Transformer layers to learn the cross-modal interaction between the words and visual regions. The components of the cross-modality Transformer (CMT) layer are the same as the Transformer.

To obtain image-aware word representations, we stack two CMT layers to perform superior-level semantic interaction. These two CMTs are internally calculated in the same way, except that the $Q$, $K$, and $V$ are from different sources. In the first stage, we first perform multi-head Cross-Model Attention by treating $v$ as $Q$, and $T$ as $K$ and $V$:

$$CA_i(v,T) = soft\max(\frac{[W_{qi}v]^T[W_{ki}T]}{\sqrt{d/m}})[W_{vi}T]^T \tag{3}$$

$$MA(v,T) = W'[CA_1(v,T),...,CA_m(v,T)]^T \tag{4}$$

where $CA_i$ is the $i-th$ head of Cross-Modal Attention, $\{W_{qi}, W_{ki}, W_{vi}\} \in \mathbb{R}^{d/m \times d}$ refers to the weight matrices for the $Q$, $K$, and $V$ respectively, and $W' \in \mathbb{R}^{d \times d}$ multi-head attention. Then, we obtain the output $P = \{p_0, p_1, ..., p_{n+1}\}$ of the first CMT layer:

$$P' = LN(v + MA(v,T)) \tag{5}$$

$$P = LN(P' + FFN(P')) \tag{6}$$

where $LN$ and $FFN$ stand layer normalization and feed-forward network respectively. In the second stage, we treat $T$ as $Q$, $P$ as $K$ and $V$. Then the second CMT layer generates the image-aware word representations $A = \{a_0, a_1, ..., a_{n+1}\}$.

To obtain word-aware visual representations, we use a CMT layer to perform basic-level semantic interaction. We treat $T$ as $Q$, and $v$ as $K$ and $V$. Then the word-aware visual representations $Q = (q_0, q_1, ..., q_{n+1})$ can be computed through Equation. 3-Equation. 6.

To trade off the cross-modality contributions, we use a gate function to obtain the final semantic interaction representation $H = \{h_0, h_1, ..., h_{n+1}\}$:

$$g = \sigma(W_a^T A + W_q^T Q) \tag{7}$$

$$H = concat(A, g \cdot Q) \tag{8}$$

where $A$ are image-aware word representations, $Q$ are word-aware visual representations, $\{W_a, W_q\} \in \mathbb{R}^{d \times d}$ refer weight matrices, and $\sigma$ stands the element-wise sigmoid function.

### 3.5 CRF Decoding Module

Conditional Random Fields (CRF) take into account the correlations between labels in neighboring positions and assign a score to the entire sequence of labels. This approach can lead to improved accuracy in sequence labeling tasks. Consequently, given a sequence $X$, all the possible label sequences y can be produced as follows:

$$P(y|X) = \frac{\prod_{i=1}^n S_i(y_{i-1}, y_i, X)}{\sum_{y' \in Y} \prod_{i=1}^n S_i(y'_{i-1}, y'_i, X)} \tag{9}$$

where $S_i(y_{i-1}, y_i, X)$ and $S_i(y'_{i-1}, y'_i, X)$ are potential functions.

### 3.6 Model Training

There are two tasks in our proposed ESPVR model: MNER, and ESI. In the training phase, we jointly train the whole model. The final training objective

function $L$ is the combination of MNER loss and ESI loss. By minimizing negative log-likelihood estimation, $L$ can be denoted as:

$$L = L_{MNER} + L_{ESI} = -(\log(P(y|X)) + \lambda \log(P(z|X)))$$
(10)

where $\lambda$ is a hyperparameter to control the contribution of the auxiliary ESI module. Here we set $\lambda$ to 0.08

## 4 Experiments

In the following section, we conduct experiments on two MNER datasets, comparing our Entity Visual Regions Positioning (ESPVR) approach with several unimodal and multimodal approaches.

### 4.1 Experiment Settings

**Datasets**: We use two publicly MNER datasets (Twitter-2015 (Zhang et al., 2018) and Twitter-2017 (Lu et al., 2018) ) to evaluate the effectiveness of our framework. Twitter-2015 and Twitter-2017 include multimodal tweets from 2014 to 2015 and from 2016 to 2017 respectively. Both datasets are composed of four types of entities: Person (PER), Location (LOC), Organization (ORG), and Miscellaneous (MISC) (In Twitter-2017, the last tag is Other. Here, we collectively refer to them as MISC). Each sample in the two datasets is composed of a pair sentence, image.

**Implementation Details**: For both datasets, we use the same hyperparameters. To compare each unimodal and multimodal method in the experiment, the maximum length of the text is set to 128 which can cover all words. For our ESPVR approach, most of the hyperparameters are set in the following aspects: The word representations C are initialized with the pre-trained BERT (bert-base-uncase) model of dimension 768 by Devlin et al. (2018), and fine-tuned during training. The visual embeddings B are initialized by Swin Transformer with the dimension of 1536. Swin Transformer is fixed during training. The Self-Attention layer has a head size of 8 and a number of 4. Additionally, the Cross-Modal Attention has feature dimensions of 512. The learning rate, the dropout rate, and the tradeoff parameter are respectively set to 1e-4, 0.4, and 0.08, which can achieve the best performance on the development set of both datasets via a small grid search over the combinations of [1e-5, 1e-4], [0.1, 0.5], and [0.05, 0.9]. We implement the proposed model with PyTorch (Paszke et al., 2019). The model is trained and tested on one Nvidia GeForce-RTX 2080 GPU with batch size 32.

### 4.2 Baseline methods

We compare our ESPVR with several baseline models for a comprehensive comparison to demonstrate the superiority of our ESPVR. For unimodal, we choose BiLSTM-CRF (Huang et al., 2015), BLSTM-CNNs-CRF (Ma and Hovy, 2016), HBiLSTM-CRF (Lample et al., 2016), BERT+softmax (Devlin et al., 2018), BERT+CRF (Devlin et al., 2018), BERT-BiLSTM-CRF (Dai et al., 2019), For multimodal, we choose GVATT-HBiLSTM-CRF (Lu et al., 2018), AdaCAN-CNN-BiLSTM-CRF (Zhang et al., 2018), GVATT-BERT-CRF (Lu et al., 2018), UMT-BERT-CRF (Yu et al., 2020), MAF (Wang et al., 2020), UMGF (Zhang et al., 2021), ITA (Wang et al., 2021), UAMNer (Liu et al., 2022a), MGCMT (Liu et al., 2022b).

### 4.3 Main Results

Following the other baselines, we employ standard precision (P), recall (R), and F1 score (F1) to evaluate the overall performance and report F1 for every single type of metric. Since the two Twitter datasets differ significantly in type distribution and data characteristics, we also conduct extensive experiments in the self-domain and cross-domain cases to demonstrate the validity and generality of our approach.

**Self-domain Scenario**. Table 1 shows the overall results of unimodal and multimodal approaches on the two benchmark Twitter MNER datasets. From the table, we have the following findings:

(1) Pre-trained model BERT is more powerful than conventional neural networks. This indicates that the pre-trained model can indeed provide abundant syntactic and semantic features. CRF considers the correlations between labels in neighborhoods and scores the whole sequence of labels. Therefore, the recent approaches are typically based on BERT-CRF.

(2) Multimodal approaches can usually perform better than their corresponding unimodal approaches approaches. By comparing all multimodal and unimodal approaches, we can find that both global images and visual information of local objects are valuable to MNER. This confirms that visual information can bring a wealth of external knowledge to the text. However, this approach does not bring very significant improvement, which demonstrates that MNER

| Modality | Methods | Twitter-2017 | | | | | | | Twitter-2015 | | | | | | |
|---|---|---|---|---|---|---|---|---|---|---|---|---|---|---|---|
| | | Single Type(F1) | | | | Overall | | | Single Type(F1) | | | | Overall | | |
| | | PER | LOC | ORG | MISC | P | R | F1 | PER | LOC | ORG | MISC | P | R | F1 |
| Text | BI-LSTM-CRF | 85.12 | 72.68 | 72.50 | 52.56 | 79.42 | 73.43 | 76.31 | 76.77 | 72.56 | 41.33 | 26.80 | 68.14 | 61.09 | 64.42 |
| | BLSTM-CNNs-CRF | 87.99 | 77.44 | 74.02 | 60.82 | 80.00 | 78.76 | 79.37 | 80.86 | 75.39 | 47.77 | 32.61 | 66.24 | 68.09 | 67.15 |
| | HBiLSTM-CRF | 87.91 | 78.57 | 76.67 | 59.32 | 82.69 | 78.16 | 80.37 | 82.34 | 76.83 | 51.59 | 32.52 | 70.32 | 68.05 | 69.17 |
| | BERT+softmax | 90.88 | 84.00 | 79.25 | 61.63 | 82.19 | 83.72 | 82.95 | 84.72 | 79.91 | 58.26 | 38.81 | 68.30 | 74.61 | 71.32 |
| | BERT+CRF | 90.06 | 81.99 | 81.83 | 63.41 | 82.98 | 84.46 | 83.71 | 84.74 | 80.51 | 60.27 | 37.29 | 69.22 | 74.59 | 71.81 |
| | BERT-BiLSTM-CRF | 90.29 | 84.55 | 80.97 | 64.85 | 83.20 | 84.68 | 83.93 | 84.32 | 79.31 | 61.66 | 37.53 | 71.03 | 73.57 | 72.27 |
| Text and Image | GVATT-HBiLSTM-CRF | 89.34 | 78.53 | 79.12 | 62.21 | 83.41 | 80.38 | 81.87 | 82.66 | 77.21 | 55.06 | 35.25 | 73.96 | 67.90 | 70.80 |
| | AdaCAN-CNN-BiLSTM-CRF | 89.63 | 77.46 | 79.24 | 62.77 | 84.16 | 80.24 | 82.15 | 81.98 | 78.95 | 53.07 | 34.02 | 72.75 | 68.74 | 70.69 |
| | GVATT-BERT-CRF | 90.94 | 83.52 | 81.91 | 62.75 | 83.64 | 84.38 | 84.01 | 84.43 | 80.87 | 59.02 | 38.14 | 69.15 | 74.46 | 71.70 |
| | AdaCAN-BERT-CRF | 90.20 | 82.97 | 82.67 | 64.83 | 85.13 | 83.20 | 84.10 | 85.02 | 81.04 | 59.02 | 38.98 | 69.34 | 75.22 | 72.16 |
| | UMT-BERT-CRF | 91.56 | 84.73 | 82.24 | 70.10 | 85.28 | 85.34 | 85.31 | 85.24 | 81.58 | 63.03 | 39.45 | 71.67 | 75.23 | 73.41 |
| | UMGF | 91.92 | 85.22 | 83.13 | 69.83 | 86.54 | 84.50 | 85.51 | 84.26 | 83.17 | 62.45 | 42.42 | 74.49 | 75.21 | 74.85 |
| | ITA | 91.40 | 84.80 | 84.00 | 68.60 | —— | —— | 85.72 | 85.60 | 82.60 | 64.40 | 44.80 | —— | —— | 75.60 |
| | UAMNer | 91.86 | 85.71 | 84.25 | 68.73 | 86.17 | 86.23 | 86.20 | 85.14 | 81.66 | 62.46 | 40.95 | 73.02 | 74.75 | 73.87 |
| | MAF | 91.51 | 85.80 | 85.10 | 68.79 | 86.13 | 86.38 | 86.25 | 84.67 | 81.18 | 63.35 | 41.82 | 71.86 | 75.10 | 73.42 |
| | MGCMT | 90.82 | 86.21 | 86.26 | 66.88 | 86.03 | 86.16 | 86.09 | 85.84 | 82.03 | 63.08 | 40.81 | 73.57 | 75.59 | 74.57 |
| | M3S | 92.73 | 84.81 | 82.49 | 69.53 | 86.93 | 85.21 | 86.06 | 86.05 | 81.32 | 62.97 | 41.36 | 74.92 | 75.14 | 75.03 |
| Ours | ESPVR | 92.73 | 84.75 | 83.82 | **70.55** | 85.70 | **87.35** | **86.52** | 85.60 | 80.48 | 59.45 | 38.08 | 71.94 | 74.17 | 73.04 |

Table 1: Results on Twitter-2017 and Twitter-2015.

still has considerable space for progress in proposing a more effective multimodal approach.

(3) Our ESPVR approach achieves state-of-the-art performance on Twitter-2017 dataset and competitive results on Twitter-2015. To position the visual regions that are most relevant to the entities in the text, we design an ESPVR module. In comparison with the existing multimodal methods, our approach outperforms the state-of-the-art MAF by 0.27 on Twitter-2017 but performs slightly worse on Twitter-2015. This is because there are many unmatched text-image pairs, and it is one direction of our future work.

**Cross-domain Scenario**. Table 2 shows a performance comparison of our ESPVR approach with UMT and UMGF in a cross-domain scenario. Twitter-2017 → Twitter-2015 denotes that the trained model on Twitter-2017 is further used to test on Twitter-2015. Similarly, Twitter-2015 → Twitter-2017 denotes that the trained model on Twitter-2015 is further used to test on Twitter-2017. From this table, we can observe that our ESPVR approach significantly outperforms UMGF by 0.38 and 0.84 on the overall F1, respectively. These results further confirm the effectiveness of our model.

| Approaches | Twitter-2017 → Twitter-2015 | | | | | | | Twitter-2015 → Twitter-2017 | | | | | | |
|---|---|---|---|---|---|---|---|---|---|---|---|---|---|---|
| | Single Type(F1) | | | | Overall | | | Single Type(F1) | | | | Overall | | |
| | PER | LOC | ORG | MISC | P | R | F1 | PER | LOC | ORG | MISC | P | R | F1 |
| UMT | 80.34 | 71.30 | 47.97 | 20.13 | 64.67 | 63.59 | 64.13 | 81.24 | 67.89 | 39.52 | 31.87 | 67.80 | 55.23 | 60.87 |
| UMGF | 79.62 | 71.94 | 49.48 | 20.24 | 67.00 | 62.81 | 64.84 | 81.83 | 72.25 | 41.20 | 32.00 | 69.88 | 56.92 | 62.74 |
| ESPVR | 79.17 | 70.63 | 48.53 | 18.20 | **67.58** | 63.03 | **65.22** | 82.47 | 70.68 | 42.92 | 32.63 | 65.77 | **61.54** | **63.58** |

Table 2: Performance comparison in the cross-task scenario.

## 4.4 Ablation Study

To show the effectiveness of each component in ESPVR, we conduct an ablation study by removing the particular component from it. Table 3 shows comparison results between the full model and its ablation methods.

| Methods | Twitter-2017 | | | | | | | Twitter-2015 | | | | | | |
|---|---|---|---|---|---|---|---|---|---|---|---|---|---|---|
| | Single Type(F1) | | | | Overall | | | Single Type(F1) | | | | Overall | | |
| | PER | LOC | ORG | MISC | P | R | F1 | PER | LOC | ORG | MISC | P | R | F1 |
| Ours | 92.73 | 84.75 | 83.82 | 70.55 | 85.70 | 87.35 | 86.52 | 85.60 | 80.48 | 59.45 | 38.08 | 71.94 | 74.17 | 73.04 |
| w/o IC | 91.81 | 84.59 | 82.86 | 68.59 | 84.86 | 86.24 | 85.55 | 84.97 | 80.30 | 60.03 | 37.38 | 71.67 | 73.77 | 72.70 |
| w/o ESPVR | 91.76 | 86.12 | 83.50 | 66.88 | 85.27 | 86.09 | 85.68 | 84.76 | 80.22 | 60.81 | 37.58 | 72.70 | 73.23 | 72.97 |
| w/o IC + ESPVR | 92.22 | 83.47 | 83.88 | 67.75 | 85.19 | 86.39 | 85.79 | 84.55 | 80.59 | 60.95 | 37.77 | 72.32 | 73.46 | 72.89 |

Table 3: Ablation Study of ESPVR.

**w/o IC**. This approach completely ignores the global information brought by image captions. We remove the image captions, resulting in reduced performance, which shows image captions can add external support for each input word.

**w/o ESPVR**. This approach completely ignores the problem of fine-grained semantic correspondence between the semantic units in the text-image pair. When we remove the ESPVR module and solely train the main MNER task, there is a noticeable decline in overall recall and F1 scores, alongside a slight improvement in overall precision. The result is consistent with our hypothesis that visual regions can provide clues for fine-grained semantic interaction.

**w/o IC + ESPVR**. This approach completely ignores global visual information and local visual information. We remove image captions and the ESPVR module, resulting in significant degradation of the performance of the model, indicating that both image captions and the ESPVR module

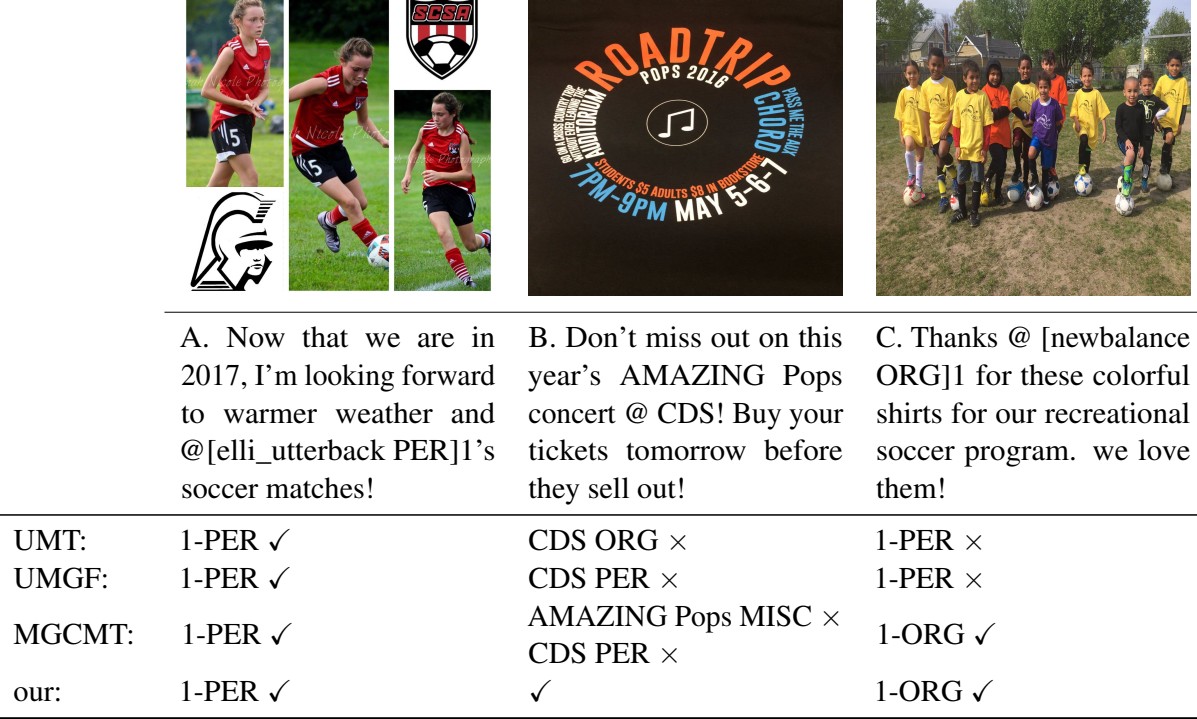

| | A. Now that we are in 2017, I'm looking forward to warmer weather and @[elli_utterback PER]1's soccer matches! | B. Don't miss out on this year's AMAZING Pops concert @ CDS! Buy your tickets tomorrow before they sell out! | C. Thanks @ [newbalance ORG]1 for these colorful shirts for our recreational soccer program. we love them! |
|---|---|---|---|
| UMT: | 1-PER ✓ | CDS ORG × | 1-PER × |
| UMGF: | 1-PER ✓ | CDS PER × | 1-PER × |
| MGCMT: | 1-PER ✓ | AMAZING Pops MISC × CDS PER × | 1-ORG ✓ |
| our: | 1-PER ✓ | ✓ | 1-ORG ✓ |

Table 4: The second row shows a few representative samples from the test set and their manually labeled entities. The bottom four rows show the prediction results of different methods for these test samples.

are essential in our framework. Removing image captions has a slightly greater impact than removing the ESPVR module. This is probably because some images are not relevant to the text. Overall, the different components of our model work effectively with each other to produce a better performance of the model in the MNER task.

### 4.5 Further Analysis

To validate the effectiveness and robustness of our method, we conduct further analysis with three specific examples in Table 4.

For informal or incomplete text, if the corresponding visual information is provided, the visual context will provide useful clues to the text. For example, in Table 4.A, the image's most obvious local visual information is the person, and all methods can obtain this local visual information with significant features. Therefore, all the multimodal approaches can correctly classify their types after incorporating the image.

It is essential to obtain local visual information from the image that is relevant to the entities in the text. If the obtained local visual information is redundant or insufficient, it may result in misidentifying a non-entity as an entity or incorrectly predicting the entity category. For example, in Table 4.B,

this text does not contain an entity, and it should not provide local visual information for this text under normal situations. However, the existing methods for obtaining local visual information all obtain a large amount of visual information from this image. So, those methods identify a non-entity as an entity. Another example is Table 4.C, UMT and UMGF use the error guidance of local visual information about the Person from the image and omit the relationship between Person and Person, resulting in the identification of "newbalance" as "PER". On the contrary, MGCMT and our method can accurately determine the entity. Here A can get the correct result because it uses the local visual information obtained in both ways, thus refining the local visual information.

### 5 Conclusion

In this paper, we present an Entity Spans Position Visual Regions (ESPVR), which obtains the most relevant visual regions for the entities in the text as fine-grained local visual information. The experimental results reveal the superiority of fine-grained local visual information acquired through this method. This information proves more advantageous in enhancing the performance of Multimodal Named Entity Recognition (MNER) compared to

using attention-based and object detection-based methods.

## Limitations

Although our experiments demonstrate the effectiveness of our method, there are still some limitations that can be improved in future work, First, data augmentation is a necessity to enhance data efficiency in deep learning. Our model lacks multimodal data enhancement. Second, there are many irrelevant text image pairs in the data of MNER, and our method aims to solve the problem of acquiring local visual information. By filtering out text-irrelevant images prior to obtaining local visual information and focusing solely on text-relevant images for acquiring local visual regions, the efficacy of our proposed method is likely to be further amplified. We hope that the insights from this work will stimulate further research on MNER performance.

## Acknowledgements

This study is in part supported by the Key Research and Development Project of Heilongjiang Province (2022ZX01A34), the 2020 Heilongjiang Province Higher Education Teaching Reform Project (SJGY 20200320).

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
