# OpenReview forum: "ESPVR: Entity Spans Position Visual Regions for Multimodal Named Entity Recognition"
_EMNLP/2023/Conference — EMNLP 2023 Findings_

### Official Review · Reviewer_Xr7Z · 2023-08-02

**Soundness:** 2

**Excitement:**

3: Ambivalent: It has merits (e.g., it reports state-of-the-art results, the idea is nice), but there are key weaknesses (e.g., it describes incremental work), and it can significantly benefit from another round of revision. However, I won't object to accepting it if my co-reviewers champion it.

**Paper Topic And Main Contributions:**

The primary objective of this paper is to explore multimodal named entity recognition, and to propose a novel approach called Entity Spans Position Visual Regions (ESPVR) that effectively captures the most relevant visual regions for entities in text. Specifically, the ESPVR module is comprised of two key components: the Entity Span Identification (ESI) module and the Visual Regions Positioning (VRP) module. The experimental results demonstrate that the proposed method outperforms current state-of-the-art techniques on the Twitter-2017 dataset, and yields competitive results on Twitter-2015.

**Questions For The Authors:**

Please refer to Reasons to reject.

**Reasons To Accept:**

The effectiveness of the proposed method is evaluated across diverse tasks, including self-domain and cross-domain scenarios, yielding promising results.

**Reasons To Reject:**

1. The motivation behind the paper is not entirely convincing, and it would be better to provide some practical examples to support the argument. Additionally, the author points out a flaw in current MNER models, stating that attention mechanisms tend to be dispersed across the entire image, making it difficult to focus on the most relevant visual regions for the text. However, it is worth noting that the proposed VRP module is also based on attention mechanisms, which presents a contradiction between the two arguments.

2. It would be more persuasive to include a visualization experiment to demonstrate the effectiveness of the proposed ESPVR module in capturing the most relevant visual parts for text entities. The current experimental design does not provide an intuitive way to showcase this.

3. I believe that this paper is not yet ready for submission. It is important to adhere to the basic standards of formal English writing and ensure that the paper is well-organized. Additionally, Figure 1 has a low resolution and contains errors, such as the misspelling of "Multimodal Inerrection Module," which should be corrected to "Multimodal Interaction Module."

**Reproducibility:**

3: Could reproduce the results with some difficulty. The settings of parameters are underspecified or subjectively determined; the training/evaluation data are not widely available.

**Reviewer Confidence:**

4: Quite sure. I tried to check the important points carefully. It's unlikely, though conceivable, that I missed something that should affect my ratings.

---

> ### Author Rebuttal · Authors · 2023-08-29
>
> Thank you very much for your efforts in improving the quality of our paper. Please allow us to address the key comments you raise:
>
> 1.The motivation behind the paper is not entirely convincing, and it would be better to provide some practical examples to support the argument. Additionally, the author points out a flaw in current MNER models, stating that attention mechanisms tend to be dispersed across the entire image, making it difficult to focus on the most relevant visual regions for the text. However, it is worth noting that the proposed VRP module is also based on attention mechanisms, which presents a contradiction between the two arguments.
>
> （1）We think this proposal is very valuable. We have found examples that support our motivation and will add to the final version of the paper. Since examples require the use of images, but here it is not possible to insert images, we are very sorry that we are not able to describe examples here.
>
> （2）Our approach doesn't contradict that viewpoint. The attention mechanism itself is not problematic and is a very good method. Although both use the attention mechanism, the local visual information we acquire is the most relevant to the entities in the text. When previous work used the attention mechanism to acquire visual information, there was no special consideration for the fact that attention would be distracted by the entire image, resulting in previous work acquiring visual information that was not only relevant to the text, but also contained a large amount of irrelevant visual information, which ultimately impacted the prediction. The essence of our method is to calculate the attention distribution of entities in the text and visual regions in the image, discard the part of the visual regions with low relevance scores through the VRP module, and keep the part of the visual regions that are most relevant to the entities in the text as the local visual information, which solves the problem of a large amount of irrelevant visual information due to the distraction problem of the attention mechanism.
>
> 2.It would be more persuasive to include a visualization experiment to demonstrate the effectiveness of the proposed ESPVR module in capturing the most relevant visual parts for text entities. The current experimental design does not provide an intuitive way to showcase this.
>
> We think this proposal is very valuable. We have obtained the visualization results and will add them to the final version of the paper. We are very sorry that we are unable to show the visualization results here, since the visualization requires the use of images, but they cannot be inserted here.
>
> 3.I believe that this paper is not yet ready for submission. It is important to adhere to the basic standards of formal English writing and ensure that the paper is well-organized. Additionally, Figure 1 has a low resolution and contains errors, such as the misspelling of "Multimodal Inerrection Module," which should be corrected to "Multimodal Interaction Module."
>
> We apologize profusely for our carelessness. We have invited professionals to polish the language in this paper. Moreover, we have carefully checked the whole text , and now there are no spelling mistakes, and we have modified Figure 1.

---

### Official Review · Reviewer_mY78 · 2023-08-07

**Soundness:** 2

**Excitement:**

3: Ambivalent: It has merits (e.g., it reports state-of-the-art results, the idea is nice), but there are key weaknesses (e.g., it describes incremental work), and it can significantly benefit from another round of revision. However, I won't object to accepting it if my co-reviewers champion it.

**Paper Topic And Main Contributions:**

This paper focuses on the task of MNER.
To solve the two drawbacks existing in previous methods, this paper proposes a novel method, called ESPVR.
The experiments on two datasets demonstrate the effectiveness of the proposed method.

**Reasons To Accept:**

- This paper is well-written and is easy to follow.

**Reasons To Reject:**

- The reported experimental results cannot strongly demonstrate the effectiveness of the proposed method.
  - In Table 1, for the proposed method, only 6 of the total 14 evaluation metrics achieve SOTA performances.
  - In Table 2, for the proposed method, only 8 of the total 14 evaluation metrics achieve SOTA performances. In addition, under the setting of "Twitter-2017 $\rightarrow$ Twitter-2015", why the proposed method achieves best overall F1, while not achieves best F1 in all single types?
  - In Table 3, for the proposed method, 9 of the total 14 evaluation metrics achieve SOTA performances, which means that when ablating some modules, the performance of the proposed method will improve. Furthermore,  The performance improvement that adding a certain module can bring is not obvious.
- In line 284, a transformer layer with self-attention is used to capture the intra-modality relation for the test modality. However, there're a lot of self-attention transformer layers in BERT. Why not using the attention scores in the last self-attention transformer layer?
- In line 322, softmmax -> softmax
- Will the coordination of $b_d$ exceed the scope of the patches?

**Reproducibility:**

4: Could mostly reproduce the results, but there may be some variation because of sample variance or minor variations in their interpretation of the protocol or method.

**Reviewer Confidence:**

3: Pretty sure, but there's a chance I missed something. Although I have a good feel for this area in general, I did not carefully check the paper's details, e.g., the math, experimental design, or novelty.

---

> ### Author Rebuttal · Authors · 2023-08-29
>
> We thank the reviewer for your valuable comments, which will serve to improve the paper considerably. Please allow us to address the key comments you raise:
>
> 1.The reported experimental results cannot strongly demonstrate the effectiveness of the proposed method.
>
> （1）The main reason for this problem is due to the presence of many irrelevant image-text pairs in the Twitter dataset [1, 2, 3, 4], especially Twitter-2015. The contribution of this paper is to firstly discover two problems with existing methods for acquiring local visual information, and propose a method. Although the effect of our proposed method does not achieve SOTA on all evaluation metrics, we still significantly outperform other works in some important metrics. This is enough to show that the motivation of the research in this paper is correct, and that the method we propose is effective. We do believe this can advance the growth of MNER.
>
> The main limitation of this paper is that it does not deal with the problem of image-text irrelevance. In our future work, we will propose an effective method to solve the problem of image-text irrelevance , by filtering out text-irrelevant images before acquiring the local visual information, and acquiring the local visual region only from the text-related images, in order to further improve the modeling effect.
>
> （2）We reran the "Twitter-2017→Twitter-2015" cross-domain experiment and the results remain the same as documented in the paper, where the evaluation metrics are also correct. To further prove the authenticity of the experiments, before 31 Aug 2023, 23:59 China Standard Time，we will put the code and data about the "Twitter-2017→Twitter-2015" and "Twitter-2015→Twitter-2017" experiments on github, and named it ESPVR.
>
> 2.In line 284, a transformer layer with self-attention is used to capture the intra-modality relation for the test modality. However, there're a lot of self-attention transformer layers in BERT. Why not using the attention scores in the last self-attention transformer layer?
>
> The self-attention effect is related to the number of layers of the transformer used, and this paper achieves the best self-attention effect with the addition of these 4 layers of the transformer.
>
> 3.In line 322, softmmax -> softmax
>
> We apologize greatly for our carelessness. We have carefully checked the entire text and there are no spelling errors at this time.
>
> 4.Will the coordination of  $b_d$ exceed the scope of the patches?
>
> This does not happen. When we wrote the algorithm, we took the out-of-bounds problem into account. The coordinates $d$ of $b_d$ are obtained by adding $\alpha$ to the horizontal and vertical coordinates of the coordinate $t$, respectively. If the horizontal coordinate of coordinate $t$ crosses the line after adding $\alpha$, the rightmost coordinate is used as the horizontal coordinate of $d$. If the vertical coordinate of coordinate $t$ crosses the line after adding $\alpha$, the bottom coordinate is used as the horizontal coordinate of $d$.

---

### Official Review · Reviewer_b5zS · 2023-08-10

**Soundness:** 3

**Excitement:**

4: Strong: This paper deepens the understanding of some phenomenon or lowers the barriers to an existing research direction.

**Paper Topic And Main Contributions:**

This work introduces a novel technique for named entity recognition (NER) that combines image and text modalities. The focal point of this work is the introduction of a novel learning algorithm module named "Entity Spans Position Visual Regions" (ESPVR), designed to address common challenges encountered in existing solutions. The authors claim that the ESPVR module enhances the efficacy of multimodal named entity recognition by identifying the most relevant visual regions corresponding to entities within the text. This information is subsequently utilized to enhance the attention layer, facilitating improved information concentration. In contrast to extant methodologies relying on the entire image feature map, the proposed ESPVR module offers a more targeted and effective approach. The research methodology is substantiated by a well-designed experimental setup, featuring both quantitative and qualitative assessments.

**Questions For The Authors:**

1. Upon the under-performance on Twitter 2015, please explain how would you move further to tackle such issue?

2. Please offer sample evidences on the claimed issue existed in the dataset of Twitter 2015?

3. Regarding the model design, do the authors posit that an alternative structure, such as a split-attention based network with segmented image subdomains. Do you think this would benefit or harm the model's performance?

**Reasons To Accept:**

First, the present work is well presented, demonstrating a clear and good motivation to address a prevailing challenge within the field. The lack of identified visual regions for NER stands as a long-existed issue in the domain. The solution proposed by authors offers a remedy to this issue. The innovative proposition of identifying the most relevant visual features based on textual cues not only shows authors' novelty but also lays a new direction for future studies.

Secondly, this work offers solid mathematical justification and derivation for the methodology and good explanation on the methodology. This improves the reproducibility of this work.

Thirdly, this study is substantiated by a comprehensive set of experiments, including good ablation studies. The results, presented both qualitatively and quantitatively, justifies the proposed model's theoretical and practical values.

**Reasons To Reject:**

The dataset used for experimentation lacks sufficient diversities, mainly single sourced on Twitter's data. However, it is possible that the inherent correlation between images posted on Twitter and their associated textual content results in a stronger coherence unique to the Twitter platform, compared to alternative data sources, such as E-commerce's data. Consequently, further evidence is needed to establish the extrapolation of the present work's scalability and efficacy to other domains.

A minor worry arises from the model's marginal enhancement and lack of progress on the Twitter 2015 dataset. While the authors briefly address reasons of solution's underperformance, they don't provide substantiating evidence. It's recommended to share their rationale and supporting data. Additionally, exploring other perspectives, like investigating potential disconnection between images and text in Twitter 2015, could provide valuable insights. All these would offer a clear understanding of the model's limitations for future studies.

**Reproducibility:**

4: Could mostly reproduce the results, but there may be some variation because of sample variance or minor variations in their interpretation of the protocol or method.

**Reviewer Confidence:**

3: Pretty sure, but there's a chance I missed something. Although I have a good feel for this area in general, I did not carefully check the paper's details, e.g., the math, experimental design, or novelty.

---

> ### Author Rebuttal · Authors · 2023-08-29
>
> We sincerely appreciate the valuable comments. Please allow us to address the key comments you raise:
>
> __Reasons To Reject:__
>
> 1.The dataset used for experimentation lacks sufficient diversities, mainly single sourced on Twitter's data. However, it is possible that the inherent correlation between images posted on Twitter and their associated textual content results in a stronger coherence unique to the Twitter platform, compared to alternative data sources, such as E-commerce's data. Consequently, further evidence is needed to establish the extrapolation of the present work's scalability and efficacy to other domains.
>
> We think this is an excellent suggestion. Therefore, we again looked at the datasets used in the previous work, and also consulted with experts in the field, and we found that the datasets currently available for English multimodal named entity recognition include Twitter-2015, Twitter-2017 and SNAP, where Twitter-2017 is the version after removing the illegal information from SNAP. Although we cannot experiment in other domains at this time, we experimented with the same datasets as the work mentioned in the comparison experiment. We will follow the development of the dataset and add relevant experiments for future work.
>
> 2.A minor worry arises from the model's marginal enhancement and lack of progress on the Twitter 2015 dataset. While the authors briefly address reasons of solution's underperformance, they don't provide substantiating evidence. It's recommended to share their rationale and supporting data. Additionally, exploring other perspectives, like investigating potential disconnection between images and text in Twitter 2015, could provide valuable insights. All these would offer a clear understanding of the model's limitations for future studies.
>
> The reasons why the model didn't work well on Twitter-2015 include two: many image-text pairs are irrelevant [1, 2, 3, 4], and many entities are mislabeled [5]. In fact, there are many such instances. We think that they have affected our MNER performance. We give relevant references, and a sample for each reason. Also, we will add the relevant samples to the final version of the paper.
>
> (1) Many image-text pairs are irrelevant.
>
> The content of the text is "Nice image of Kevin Love and Kye KorveDuring 1st half #NBAFinals #Cavsin9 #Cleveland". Obviously, The text talks about basketball players, while the image depicts two bins standing in front of a wall.
>
> (2) Many entities are mislabeled.
>
> The content of the text is "Dr. Julie Hood, right, teacher w/Dade schools, w/fiance Marilyn. They're getting married in a few weeks.". In the dataset, "Julie Hood" is labeled "I-PER I-PER", but the correct label for "Julie Hood" is "B-PER I-PER".
>
> [1] Vempala A, Preoţiuc-Pietro D. Categorizing and inferring the relationship between the text and image of twitter posts[C]//Proceedings of the 57th annual meeting of the Association for Computational Linguistics. 2019: 2830-2840.
>
> [2] Yu J, Jiang J, Yang L, et al. Improving multimodal named entity recognition via entity span detection with unified multimodal transformer[C]. Association for Computational Linguistics, 2020.
>
> [3] Chen D, Li Z, Gu B, et al. Multimodal named entity recognition with image attributes and image knowledge[C]//Database Systems for Advanced Applications: 26th International Conference, DASFAA 2021, Taipei, Taiwan, April 11–14, 2021, Proceedings, Part II 26. Springer International Publishing, 2021: 186-201.
>
> [4] Liu L, Wang M, Zhang M, et al. Uamner: uncertainty-aware multimodal named entity recognition in social media posts[J]. Applied Intelligence, 2022, 52(4): 4109-4125.
>
> [5] Liu P, Wang G, Li H, et al. Multi-Granularity Cross-Modality Representation Learning for Named Entity Recognition on Social Media[J]. arXiv preprint arXiv:2210.14163, 2022.
>
> __Questions For The Authors:__
>
> 1.Upon the under-performance on Twitter 2015, please explain how would you move further to tackle such issue?
>
> In future work, we will propose an effective method to solve the problem of image-text irrelevance, by filtering out text-irrelevant images before acquiring the local visual information, and acquiring the local visual region only from the text-relevant images. Since the problem of image-text irrelevance exists not only in Twitter 2015, but also has a small portion in Twitter 2017, we believe that this method will achieve better results on both datasets.
>
> 2.Please offer sample evidences on the claimed issue existed in the dataset of Twitter 2015?
>
> The answer to this question can be found in response to question 2 in "Reasons To Reject".
>
> 3.Regarding the model design, do the authors posit that an alternative structure, such as a split-attention based network with segmented image subdomains. Do you think this would benefit or harm the model's performance?
>
> The use of Swin Transformer is part of our method, and replacing it with a split-attention based network will have a little impact on our method, but not much. Swin Transformer can split the image into many patches, and eventually we can get the visual feature representation corresponding to each pacth. The ESPVR module consists of two modules: ESI module, VRP module. Firstly, the ESI module identifies all entity spans in the text, then the VRP module uses the entity spans to obtain entity features and uses the entity features to locate the visual regions. In the step of localizing the visual visual region, we do the computation through the entity features and the corresponding visual feature of the pacth, and select the patches that are most relevant to the entity to form the local visual information. In other words, Swin Transformer can get the visual feature of each patch, which is helpful for us to locate the local visual information that is most relevant to the entities in the text.

---

### Official Review · Reviewer_YD3t · 2023-08-12

**Soundness:** 3

**Excitement:**

4: Strong: This paper deepens the understanding of some phenomenon or lowers the barriers to an existing research direction.

**Paper Topic And Main Contributions:**

The paper is motivated by the challenges and shortcomings of existing methods for acquiring visual information in the context of Multimodal Named Entity Recognition (MNER). Specifically, the authors identify two main drawbacks:

1) Attention-based methods that extract visual regions related to the text are distracted by the entire image, rather than focusing on the most relevant visual regions.
2) Object detection methods, such as Mask R-CNN, have a limited range of recognition categories and may produce visual regions that do not correspond to entities in the text.

Motivated by the above challenges, this paper presents a novel model, ESPVR, for the task of Multimodal Named Entity Recognition (MNER), which extracts entities from text and classifies them with the assistance of image information. The ESPVR model consists of four major modules: Feature Extraction, Entity Spans Position Visual Regions (ESPVR), Multimodal Interaction, and CRF Decoding. It employs BERT for text encoding and Swin Transformer for visual representation

Experiments show that the ESPVR achieved SOTA in both Twitter datasets.

**Questions For The Authors:**

1. Since ESPVR is jointly trained by ESI loss and MNER loss, it would be interesting to see qualitative/quantitative analysis about how ESPVR improve ESI task.

**Reasons To Accept:**

1. The motivation of this paper is clear.

2. The proposed ESPVR model has a rich architecture that creatively integrates visual and textual information. The combination of BERT for text and Swin Transformer for visual representation is novel and could be promising.

3. The idea is innovative and well-articulated.

**Reasons To Reject:**

1. There can be more details for implementation details. e.g. What is the lamda value at row 423? Is it learnable or manually selected? In this joint learning task, which loss plays a more important role for training?

**Reproducibility:**

4: Could mostly reproduce the results, but there may be some variation because of sample variance or minor variations in their interpretation of the protocol or method.

**Reviewer Confidence:**

2: Willing to defend my evaluation, but it is fairly likely that I missed some details, didn't understand some central points, or can't be sure about the novelty of the work.

---

> ### Author Rebuttal · Authors · 2023-08-29
>
> We feel great thanks for your professional review work on our article. Please allow us to address the key comments you raise:
>
> __Reasons To Reject:__
>
> 1.There can be more details for implementation details. e.g. What is the lamda value at row 423? Is it learnable or manually selected? In this joint learning task, which loss plays a more important role for training?
>
> The lamda value in line 423 is 0.08 and was chosen manually. Because the main task is modeled larger, the loss of the main task MNER is more useful for training in joint learning.
>
> __Questions For The Authors:__
>
> 1.Since ESPVR is jointly trained by ESI loss and MNER loss, it would be interesting to see qualitative/quantitative analysis about how ESPVR improve ESI task.
>
> The effect of the ESI task during joint training was as follows:
>
> Twitter-2017: 90.38(P), 92.38(R), 91.37(F1)
>
> Twitter-2015: 81.49(P), 82.97(R), 82.22(F1)
>
> From the experimental results, the effect of the auxiliary task ESI exceeded the effect of the primary task MNER, indicating that the auxiliary task was favorable to the primary task, and that the primary task did not impair the effect of the auxiliary task. We will add that part of the experiment and its analysis to the paper later.

---

### Meta-Review · Area_Chair_SXF3 · 2023-09-07

**Recommendation:** 3

**Metareview:**

This work introduces a novel technique for NER augmented with the visual modality, and introduces a novel learning algorithm module named Entity Spans Position Visual Regions (ESPVR), designed to address common challenges encountered in existing solutions. The authors claim that the ESPVR module enhances the efficacy of multimodal named entity recognition by identifying the most relevant visual regions corresponding to entities within the text. The reviewers generally agree on the novelty of the work and soundness of the presented method. There has been minor concern on missing experimental setup details, persuasiveness of some results, and sufficient diversity in experimental datasets.

---

### Decision · Program_Chairs · 2023-10-07

**Decision:**

Accept-Findings

**Comment:**

This work introduces a novel technique for NER augmented with the visual modality, and introduces a novel learning algorithm module named Entity Spans Position Visual Regions (ESPVR), designed to address common challenges encountered in existing solutions. The authors claim that the ESPVR module enhances the efficacy of multimodal named entity recognition by identifying the most relevant visual regions corresponding to entities within the text. The reviewers generally agree on the novelty of the work and soundness of the presented method. There has been minor concern on missing experimental setup details, persuasiveness of some results, and sufficient diversity in experimental datasets.